# Radiotherapy of Orbital and Ocular Adnexa Lymphoma: Literature Review and University of Catania Experience

**DOI:** 10.3390/cancers15245782

**Published:** 2023-12-10

**Authors:** Madalina La Rocca, Barbara Francesca Leonardi, Maria Chiara Lo Greco, Giorgia Marano, Irene Finocchiaro, Arianna Iudica, Roberto Milazzotto, Rocco Luca Emanuele Liardo, Viviana Anna La Monaca, Vincenzo Salamone, Antonio Basile, Pietro Valerio Foti, Stefano Palmucci, Emanuele David, Silvana Parisi, Antonio Pontoriero, Stefano Pergolizzi, Corrado Spatola

**Affiliations:** 1Radiation Oncology Unit, Department of Biomedical, Dental and Morphological and Functional Imaging Sciences, University of Messina, 98122 Messina, Italy; bf.leonardi@outlook.it (B.F.L.); mariachiaralg@gmail.com (M.C.L.G.); giorgiamarano@gmail.com (G.M.); irene.finocchiaro.com@gmail.com (I.F.); ariannaiudicaa@gmail.com (A.I.); silvana.parisi@unime.it (S.P.); apontoriero@unime.it (A.P.); stefano.pergolizzi@unime.it (S.P.); 2Radiation Oncology Unit, Department of Medical Surgical Sciences and Advanced Technologies “G.F. Ingrassia”, University of Catania, 95123 Catania, Italy; r.milazzotto@policlinico.unict.it (R.M.); lucaliardo@hotmail.com (R.L.E.L.); vivianalamonaca@gmail.com (V.A.L.M.); v.salamone966@gmail.com (V.S.); 3Department of Medical Surgical Sciences and Advanced Technologies “GF Ingrassia”, University Hospital Policlinico “G. Rodolico-San Marco”, 95123 Catania, Italy; basile.antonello73@gmail.com (A.B.); pietrofoti@hotmail.com (P.V.F.); spalmucci@unict.it (S.P.); 4Radiology I Unit, Department of Medical Surgical Sciences and Advanced Technologies "G.F. Ingrassia", University of Catania, 95123 Catania, Italy; david.emanuele@yahoo.it

**Keywords:** orbit, orbital tumors, lymphoma, ocular oncology, radiation oncology, radiotherapy, proton beam radiotherapy

## Abstract

**Simple Summary:**

Lymphomas are rarely localized in the ocular and orbital regions. Although their diagnosis is often complicated due to the similarity with other pathologies of this anatomical district, they are characterized by a good prognosis in the vast majority of cases. This article reports our clinical experience in the field, associating it with a review of the main therapeutic options, placing greater attention on the role of radiotherapy in the treatment of these tumors.

**Abstract:**

Orbital and ocular adnexa lymphomas are rare neoplasms confined to the orbital region. The prognosis is generally favorable, with a high proportion of localized disease, indolent clinical course, prolonged disease-free intervals, and low lymphoma-related mortality rate. We report our experience on eleven patients with confirmed histological diagnosis of lymphoma stage IE-IIE, treated between 2010 and 2021 with radiotherapy alone or in association with chemotherapy or immunotherapy. Eight patients were treated with primary radiotherapy only, while three received previous systemic treatments. Six patients were treated with Proton beam therapy (PBT), and five with external beam radiotherapy (EBRT). The five-year local control rate was 98%; only one patient developed an out-of-field recurrence. We also conducted a comprehensive literature review using electronic databases (PubMed, EMBASE, and Cochrane Library). Articles were selected based on their pertinence to treatment of the ocular and adnexal lymphoma focusing on radiotherapy techniques (electron beam radiotherapy, photon beam radiotherapy, or proton beam radiotherapy), treatment total dose, fractionation schedule, early and late radio-induced toxicities, and patient’s clinical outcome. Radiotherapy is an effective treatment option for orbital lymphoma, especially as standard treatment in the early stage of orbital lymphoma, with excellent local control rate and low rates of toxicity.

## 1. Introduction

Orbital and ocular adnexa lymphomas (OOAL) are a rare localization of lymphomas confined to the orbital region, accounting for approximately 7–8% of all non-Hodgkin’s lymphomas (NHLs) [1,2]. These neoplasms develop from B-lymphocytes, T-lymphocytes, or NK lymphocytic cells [3,4]. It is recognizable by its morphological heterogeneous appearance with variable presence of germinal centers, plasma cells, and/or monocytic lymphocytes [4]. Within the NHLs, histological subtypes can be divided into indolent or low-grade lymphoma (i.e., extranodal marginal zone lymphoma (EMZL)) involving mucosa-associated lymphoid tissue (MALT), follicular lymphoma (FL) or lymphoplasmacytic lymphoma) and aggressive or high-grade lymphomas (e.g., diffuse large B-cell lymphoma (DLBCL) or mantle-cell lymphoma (MCL)) with different patterns of response to local and systemic treatment, local or distant recurrence [5,6]. An 80% of lymphomas involving ocular adnexa (conjunctiva, orbit, lacrimal gland, and eyelid) have mature B-cell origin [4] as a response to persistent antigenic stimulation in case of chronic inflammatory or autoimmune disorders [7]. Association with *C. psittaci* infection is also documented, especially in Eastern nations, even if there is no overall acceptance [8,9]. These types of lesions must be distinguished from intraocular lymphoma, a subtype of primary central nervous system lymphoma that represents a high-grade process with a distinct therapeutic strategy, but is not the subject of this review [10]. Lymphoma occurs predominantly in elder individuals [11] with an incidence peak amongst the fifth and seventh decade of life (median age ∼65 years), with a female predominance (male/female = 1:1.5/2) in the Western population. In contrast, a Korean study by Cho et al. revealed a significantly younger age at diagnosis (median age of 46 y/o), with a male predominance [11]. Ucgul et al. reported the rarity of this pathology in the pediatric population [12]. Treatment may include radiotherapy (RT), chemotherapy (CHT), immunomodulating therapy, primary antibiotic treatment, surgical excision, or combination therapy. The choice of treatment depends on a variety of factors: histopathologic type, systemic staging at the initial presentation, and patient comorbidities. When in the early stage, radiotherapy alone is the curative option in patients diagnosed with indolent lymphoma to achieve high response rates with manageable toxicity [13,14]. In this article, we report our clinical experience in addition to a review of the literature, focusing on the use of radiotherapy (external beams and protons) and the different therapeutical doses in the treatment of OOAL. We want to draw attention to the primary role of radiotherapy, in association with systemic therapy in the most aggressive histologies or as the only treatment, with long-lasting and stable loco-regional control, especially in case of indolent disease. The few and manageable side effects allow access to radiation treatment for a larger number of patients (e.g., elderly, patients with multiple comorbidities, autoimmune disorders, etc.).

## 2. Materials and Methods

We retrospectively investigated eleven patients, six women and five men, with confirmed histological diagnosis of MALT lymphoma (nine of them) and follicular lymphoma (FL) (two of them), treated in our center between 2010 and 2021. Median age at the diagnosis was 59 years (range 31–75). The pretreatment staging included: an ophthalmological examination, a full blood examination (complete blood count, liver function tests, and lactate dehydrogenase (LDH)), magnetic resonance imaging (MRI) of the orbit, chest, abdomen, and pelvis, computed tomography (CT) and positron-emission tomography (PET). We classified the population using the Ann Arbor Staging System: eight patients with stage IE and three with IIE disease. Eight of them were treated with primary radiotherapy only, three received previous systemic treatments. These three patients came to our observation only after having undergone systemic therapy at other institutions. Six patients were treated with proton beam therapy (PBT), and five with external beam radiotherapy (EBRT). The mean radiation dose was 35 Gy (range 20–39,6 Gy). Population characteristics are summarized in detail in Table 1. Toxicities were graded according to CTCAE version 5.0 [15]. We also conducted a comprehensive literature review using electronic databases (PubMed, EMBASE, and Cochrane Library). Articles were selected based on their pertinence to treatment of the ocular and adnexal lymphoma focusing on radiotherapy management. We selected studies performed between 2001 and 2022, using the following keywords: “ocular adnexa”, “orbital lymphoma”, “radiotherapy”, “ocular lymphoma”, “orbital lymphoma”, “MALT lymphoma”; “conjunctiva”; “orbit”; “non-Hodgkin lymphoma”, “periorbital lymphoma”. We focused our review on treatment strategies, with particular interest on radiotherapy techniques (electron beam radiotherapy, photon beam radiotherapy, or proton beam radiotherapy), treatment total dose, fractionation schedule, early and late radio-induced toxicities, and clinical outcome. 

## 3. Clinical Presentation

OOALs are often insidious, due to few and aspecific symptoms such as localized pain, conjunctival swelling, redness, and itching irritation, leading to a differential diagnosis with other benign pathologies, often masqueraded by the use of topical steroids, that can delay diagnosis. It has been estimated a median interval between onset symptoms and definitive diagnosis of 7 months ranging from 1 month up to 10 years [7]. Clinical presentations may vary depending on the primary localization. For example, if lymphomatous lesions are localized on the eyelid, a palpable rubbery or firm mass can be observed with other compression symptoms such as periorbital edema, decreased visual acuity, and motility disorders. Furthermore, progressive proptosis and diplopia can be objective if the dermis or the orbicular muscles are involved. Conjunctival involvement is characterized by a salmon-pink-colored lesion with slow infiltrative growth; other uncommon symptoms are conjunctival hyperemia, blurry vision, chemosis, ectropion, pterygium, photophobia, and corneal symptoms [16,17]. Lacrimal gland infiltration symptoms are dryness, pain, lacrimal duct discharge, epiphora, displacement of the globe, proptosis, and ocular motility reduction [18].

## 4. Diagnosis and Staging

The initial evaluation of patients is a milestone. It is important to define if the neoplasm has systemic extension or is locally confined. A complete ophthalmologic examination followed by an adequate tissue sampling for histopathologic diagnosis is mandatory [19,20]. Staging workup must include: physical examination, bone marrow biopsy, MRI, CT of head and neck, chest, abdomen, and pelvis, and 18-fluorodeoxyglucose-positron emission tomography-CT (FDG-PET-CT) to complete staging of the patient [20,21]. CT and MRI studies of the facial district are important to evaluate localization, size, and local infiltration of the mass, which helps identify the correct staging and also the following treatment strategy (i.e., choosing between intralesional or systemic therapy, radiotherapy planning, etc.). In head CT scanning, OOALs usually present as a circumscribed lesion with higher density (in comparison with the brain tissue) that tends to infiltrate nearby tissues and show moderate enhancement when contrast is used. It is also possible to detect bone erosion and calcifications. MRI provides complementary information and demonstrates extension into the adjacent brain, muscles, or sinuses; in particular, hyperintense tissue in T2-weighted sequences are often hypercellular masses [22,23]. The clinical stage of OOAL is determined by the Ann Arbor staging classification and the American Joint Committee on Cancer Tumor, Node, Metastasis (TNM), as shown in Table 2 and Table 3 [24,25,26]. Approximately 85–90% of patients with a diagnosis of OOAL have stage I disease; nodal involvement is reported in 5% of patients; only 10–15% of patients have disseminated disease [7].

## 5. Treatment Options

Various strategies of treatment are available depending on the initial stage of the neoplasm: surgical excision, radiotherapy, chemotherapy, anti-chlamydial antibiotic (doxycycline and clarithromycin) [7,9], and immunomodulating therapy or combination therapy. Tailoring the treatment requires a multidisciplinary approach which should consider extensions, patient comorbidities, disease-related prognostic factors, short and long-term efficacy, and toxicities impact on a patient’s quality of life [7]. Radiotherapy alone is usually the treatment of choice, in early stage and indolent OOAL (stage I–IIE) [27]. Patients with stage III or IV or with aggressive OOAL’s histotypes need a systemic treatment chemotherapy/immunotherapy combined with local radiotherapy [28]. However, no prospective clinical trials have been conducted to evaluate these therapeutic options or define the optimal treatment approach for these patients. Radiotherapy and non-radiotherapy therapeutic options are summarized in detail in Figure 1.

### 5.1. Non-Radiotherapy Treatment Options

“Watch and wait” is not recommended due to the high incidence of local and systemic recurrence, but it can be taken under consideration in the setting of frail elderly patients with low-grade, asymptomatic, and unilateral disease with severe comorbidities that preclude other and more aggressive therapeutic approaches [29,30]. Surgical resection may be used to treat small or encapsulated tumors (e.g., conjunctival and lacrimal gland tumors). However, surgery alone showed a high risk of recurrence if not associated with adjuvant chemotherapy or radiotherapy [31]. In case of aggressive histological subtypes (i.e., MCL, DLBCL, T-cell lymphoma), bilateral disease, high-risk of local or distant relapse, chemotherapy regimen containing cyclophosphamide (CHOP (cyclophosphamide, adriamycin, vincristine, prednisone), COP/CVP (cyclophosphamide, vincristine, prednisone), C-MOPP (cyclophosphamide, vincristine, procarbazine, prednisone), and schemes containing Chlorambucil or Bendamustine can be taken under consideration with or without radiotherapy [32,33,34,35]. If infection with *C. psittaci* is detected, especially in Eastern countries, antibiotics, such as Doxycycline, can be used [7,8,9]. Interferon-alpha (INF-α) is an option for OOAL localized in the conjunctiva or the lacrimal glands. This glycoprotein has already been used in other neoplasm treatments, which possess antiviral, antiproliferative, and immunomodulatory functions (i.e., apoptosis, producing tumor suppressor gene p53, inhibition of tumoral neo angiogenesis). Cellini et al. and Holds et al. documented the successful use of intralesional INF-α in the treatment of conjunctival lymphoma [36,37]. Generally, 1 million IU of INF-α are administrated weekly, with optimal local control rate. INF-α showed low toxicity, and in general, no significant local side effects after injection. Rituximab is a monoclonal chimeric anti-CD20 antibody that targets the surface antigen CD20, overexpressed on CD20-positive NHL B cells. It can be administered intralesionally in case of recurrence and local relapse [38] or intravenously for bilateral or systemic involvement [39]. As previously mentioned, it can be used alone or in combination with other chemotherapy drugs. Monoclonal antibodies can be used to deliver radioisotopes to the site of OOAL [7], with the same or better outcome compared to rituximab. Esmaeli et al., in their study, promoted a protocol where Yttrium 90-ibritumomab tiuxetan (Zevalin^®^, IDEC Pharmaceuticals Corporation, San Diego, California), a radiolabeled anti-CD20 monoclonal antibody, is administrated after 2 weekly intravenous rituximab 250 mg/mq [40].

### 5.2. Radiotherapy Treatment Options

OOALs are characterized by high radiosensitivity [41,42]. External beam radiation therapy (EBRT) is the gold standard treatment in OOAL classified as Ann Arbor Stage IE-IIE thanks to high response rates and outstanding local control [16]. Radiotherapy can be administrated as a solo treatment, with adjuvant systemic therapy, or as salvage therapy after partial or incomplete response or in case of relapse [43]. Clinical exams to evaluate the extension of orbital disease is utilized for radiotherapy plans. A review of a CT scans, MRI and/or PET scans are utilized when available in order to create a deformable fusion. The gross tumor volume (GTV) includes the tumor extent, and the clinical target volume (CTV) includes the entire GTV plus an adequate margin to fully encompass the anatomical region involving the mass. The planning target volume (PTV) includes CTV with a 10 mm to 15 mm margin [42,44]. A bolus is positioned on the skin to facilitate the superficialization of the dose, especially when palpebral skin is involved. Different particles can be adapted to deliver the dose: electrons, photons, and protons. An electron beam can be used for superficial lymphomas of the eyelid and/or conjunctiva that do not infiltrate the ocular bulb. Electron energies typically cover the entire orbit. For patients with superficial disease limited to the eyelids or conjunctiva, treatment is generally delivered using electron beams (6–12 MeV) with or without bolus to provide optimal surface dose coverage. Deeper lesions are generally treated with higher energy electron beams (9–16 MeV) or photon beams (4–6 MV) [19,45]. The photon beam has been used widely for deeper lesions involving lacrimal glands, orbital soft tissue, muscles, or for bulky mass that extends beyond the globe equator. Proton beam therapy (PBT) is a valid option for the treatment of most ocular and periorbital malignancies (e.g., melanomas, lymphomas, squamous cell carcinomas, rhabdomyosarcoma hemangioma, metastases of solid tumors) with excellent control rates [45,46,47,48,49]. Compared with conventional photon radiotherapy, PBT uses heavier subatomic particles to deliver energy with higher accuracy, conformal treatment isodoses without dose fall-off and with less scattering to the healthy surrounding tissues. Proton beams are generated by a cyclotron or synchrotron and then accelerated to proper target energy [50,51]. The proton dose distribution that may be achieved is generally superior to the dose distribution of conventional photon radiotherapy. PBT may improve the survival rate of patients by improving the local tumor treatment rate, sparing adjacent tissues and lowering collateral damage. Cirrone et al. performed a study on 4 patients with orbital non-Hodgkin lymphoma ranging between 30 and 48 Gy [RBE], with 4 fractions on 4 consecutive days [47]. Nonetheless, due to the high treatment costs for facility building and maintenance, PBT is not very diffuse. Despite the consensus on the primary role of radiotherapy, there is no universally accepted optimal radiation dose and fractionation for patients with OOAL [27]. Historically, patients have received treatment with conventional doses of 24–36 Gy with local control rates >95% and a minimum dose required of 25 Gy [6,28,52,53] Le et al. found no differences in terms of distant progression-free survival (dPFS) or local recurrence (LR) after EBRT in pts with ≤34 Gy compared with higher doses; the only difference was the increase in local toxicities observed [53]. Bhatia et al. [54] and Letschert et al. [55] treated low-grade with a median dose of 30 Gy and intermediate to high-grade tumors with a median dose of 40 Gy.

## 6. Treatment Toxicity

Treatment-independent factors, especially in elderly patients, must be considered (i.e., diabetes, glaucoma, sicca syndrome, Sjögren syndrome) because they may influence cataract development. Most patients experienced mild acute toxicities, such as skin erythema in the periorbital area, conjunctival hyperemia, chemosis, swelling, and excessive tearing, during or immediately after the radiotherapy course, which were mainly self-limited or responded to a short course of lubricant gels or skin moisturizers. These symptoms generally resolve in a few months after completion of radiotherapy [19,43]. Long-term toxicities include: radiation-induced cataracts, persistent dry eyes, and iris neovascularization [56,57]. Serious toxicities such as corneal or skin ulceration, telangiectasia, neovascular glaucoma, and radiation retinopathy are extremely rare [58,59]. Radiation-related severe late side-effects incidence increases when patients are treated with RT doses ≥35 Gy, resulting in considerable late toxic effects, including keratitis, severe dry eye syndrome, glaucoma, retinopathy, and cataract formation [28,60,61]. Stafford et al. reported a 52% rate of acute complications with doses ranging from 19 to 48 Gy [60]. Retinal toxicity may result in visual impairment, which can significantly impact quality of life [62,63]. During treatment, shielding, such as a contact lens under the eyelid, can lower the dose for healthy and sensible organs [53,64].

## 7. Discussion

External beam radiation therapy (EBRT) has been considered the standard treatment for low-grade, isolated OOALs for the past decades. Five-year local control rates with radiotherapy alone in the treatment of OAL range from 89 to 100%, as reported by several studies [6,16,21,31,42,43,44]. As previously mentioned, the pivotal role of radiotherapy as a primary treatment remains undisputed, even if there is no unanimous consensus on the ideal dose to use as a standard of treatment. In addition to the total dose and its fractionation, there are various options also depending on the type of particles used in the treatment (electrons, photons, protons) that can be considered, depending on multiple factors (e.g., extension, infiltration, lesion depth, OARs, patient comorbidities, difficulty in positioning, etc.). In our clinical experience, we treated eleven patients (mean age 59 years), and we found no significant differences in incidence related to the sex of patients forming the sample (six women and five men). We used lens shielding in patients treated with higher doses. We used both EBRT and PBT. Mean radiation dose administered was 35 Gy, ranging from 20 Gy to 39,6 Gy. Median follow-up was 9 years. Acute toxicity was mild, requiring minimal intervention: 85% experienced dry eyes, G1 (treated with eye lubricants), 70% acute blurred vision, and 15% cataracts, G2. OS, LR, and dPFS were similar in patients treated with EBRT and PBT. Lens shielding reduced the incidence of lens complications (only one patient showed Grade 3 toxicity). No cases of retinopathy or optic nerve injury were reported. The five-year local control rate was 98% and only one patient developed an out-of-field recurrence. Despite a relatively small sample, we confronted our experience with studies and clinical investigations with at least a 1-year follow-up; further details on included studies are shown in Table 4. Goda et al. endorsed the use of lens shielding in the prevention of radio-induced cataracts [43]. Taking advantage of the radiosensitivity of indolent OOAL, there has been a recent trend towards lowering radiation doses obtaining promising results of local control rates with minimal toxicity [28]. Ultra-low-dose radiotherapy (ULDRT), also called “boom-boom therapy”, demonstrated considerable overall response rates to doses of 4 Gy, administrated in 2 consecutive fractions of 2 Gy [43,44] and can be safely applied in patients with autoimmune disease, such as Sjögren syndrome [65]. Ganem et al. first reported, in a palliative setting, a ULDRT as low as 4 Gy, administered in successive fractions of 2 Gy with a high response rate of 89% [66]. Subsequently, additional clinical studies showed an acceptable clinical schedule of 4 Gy (2 Gy × 2 fractions) [20,63,67,68,69,70,71,72]. FoRT is a UK phase 3 multicentric, randomized, non-inferiority trial performed on a population of 548 patients with histological diagnosis of FL and MALT. In this trial, a conventional radiation scheme of 24 Gy in 12 fractions was compared to ULDRT with 4 Gy in 2 fractions [73]. Despite high response rates and without difference in overall survival in both arms, at a median follow-up of 22 months, the local control rate in the group that received 4 Gy was inferior to that of the group that received 24 Gy (80.4% vs 93.7%). These data are confirmed by König et al.’s [28] double-arm study. Recently, Park et al. conducted a single-center prospective phase II trial called FORMAL on fourteen patients diagnosed with stage I orbital MALToma using LDRT of 4 Gy (2 Gy × 2 fractions). They strictly monitored clinical response, giving those who did not achieve remission an additional irradiation with 24 Gy in 12 fractions [21]. Pinnix et al. [65] highlighted the importance of a correct follow-up timing in patients treated ULDRT, before other therapeutical strategies (systemic and/or reirradiation). In their series, four patients treated with ultra-low dose radiotherapy did not show CR at the first follow-up visit (3–4 months after treatment), and it was only later around 10 months that CR was achieved. More recently, Shelukar et al. [74] and de Castro et al. [75] confirmed the safety, feasibility and high local and distant control of ULDRT. Treatment with ULDRT, in patients not fit for “standard dose”, allows treatment even in patients burdened by significant comorbidities or very elderly, allowing a better quality of life and local regional control compared to observation alone. Another benefit is the possibility of re-treating or adding further fractions, if during follow-up an incomplete response or disease recurrence is found.

### Limitations of the Study

Because of the relatively low incidence of these neoplasms, this study has potential limitations. First, we performed a retrospective analysis of clinical records of a small sample: eleven patients with different neoplasm sites (five eyelids, three conjunctiva, and three lacrimal glands). Second, different treatment strategies and different dose prescriptions were performed (EBRT, PBT); our intent was to report the various options that can be used in clinical practice depending on the disease’s site, size and proximity to OARs. Third, no bioptic assessment for *C. psittaci* infection was performed on our patients. 

## 8. Conclusions

We have experienced a rising incidence of OOAL worldwide, and primary radiotherapy of indolent lymphomas has shown a high response rate and optimal local control. There is no unanimous consensus about standard fractioning or dose, even if the most accepted is conventional radiation therapy (30 Gy) for localized disease with excellent local control and combination chemotherapy or immunotherapy for disseminated disease. Furthermore, ULDRT is a convenient and effective curative option for OOAL stage I–IIE, with high response rates and durable local control in combination with close in-time follow-up with the option of reirradiation in case of locoregional relapse. Nevertheless, further studies with a larger population and long-term follow-up are needed to prove whether ULDRT may also be considered in the curative setting with overlapping local control rates and disease-free survival of CRT.

## Figures and Tables

**Figure 1 cancers-15-05782-f001:**
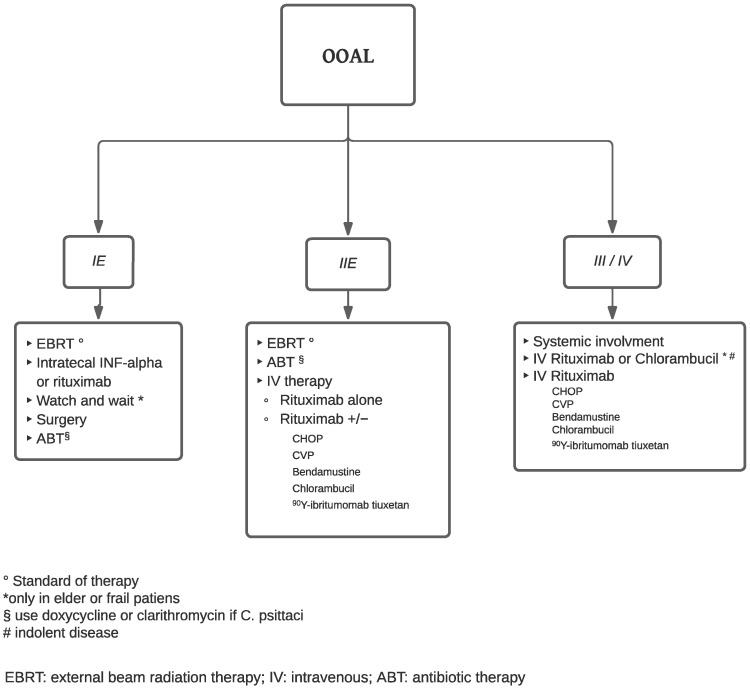
Flow chart of treatment.

**Table 1 cancers-15-05782-t001:** Patient characteristics, treatment, and outcome.

**Patients**
Male	5
Female	6
Total	11
Median age	59 (Range 31–75)
**Involved site**
Right	4
Left	7

Conjunctiva	3
Lacrimal Gland	3
Upper eyelid	3
Lower eyelid	2
**Stage**	
IE	8
IIE	3
**Hystology**	
MALT Follicular lymphoma	92
**Systemic Treatment**
None	8
Yes	3
INF	1
CHOP	1
Ab monoclonal	1
**Treatment Characteristics**
Beam Energy
X/6MV	5
p+/62MeV	6
Total Dose (No. of Fractions/Dose)
39.6 Gy (22/1.8 Gy)	2
36 GyE (4/9 GyE)	3
35 Gy (14/2.5 Gy)	1
30 GyE (3/10 GyE)	3
20 Gy (10/2Gy)	2
**Patient Outcomes**
Complete response	11
Distant relapse	1

Note: Proton therapy is indicated in Table 1 as p+. Ten patients have a histological diagnosis of MALT lymphoma; two of follicular lymphoma.

**Table 2 cancers-15-05782-t002:** Ann Arbor staging [25,26].

Ann Arbor Staging	
** *STAGE I* **	Involvement of a single lymph node region or extra lymphatic site (IE)
** *STAGE II* **	Involvement of 2 or more lymph nodes, lymphatic structures, or extra lymphatic regions alone on the same side of the diaphragm (IIE)
** *STAGE III* **	Involvement of lymph nodes on both sides of the diaphragm with localized extra lymphatic (IIIE) or splenic (IIIS) involvement, or both (IIIES)
** *STAGE IV* **	Involvement of one or more organs or tissues outside the lymphatic system
** *A: * ** *Without B symptoms* ** *B: * ** *Fever, night sweats, weight loss of >10% body weight over the last 6 months*	

**Table 3 cancers-15-05782-t003:** TNM Staging system AJCC 8th edition of OOAL [24].

	TNM Staging system AJCC 8th edition of OOAL
TX	Lymphoma extent not specified
T0	No evidence of lymphoma
T1	Lymphoma involving the conjunctiva alone without orbital involvement
T2	Lymphoma with orbital involvement ± any conjunctival involvement
T3	Lymphoma with preseptal eyelid involvement ± orbital involvement ± any conjunctival involvement
T4	Orbital adnexal lymphoma extending beyond orbit to adjacent structures, such as bone, maxillofacial sinuses and brain
NX	Involvement of lymph nodes not assessed
N0	No evidence of lymph node involvement
N1	Involvement of ipsilateral regional lymph nodes draining the ocular adnexal structure and superior to the mediastinum (preauricular, parotid, submandibular, and cervical nodes) N1a: involvement of a single lymph node region superior to the mediastinum N1b: involvement of two or more lymph node regions, superior to the mediastinum
N2	Involvement of lymph nodes regions of the mediastinum
N3	Diffuse or disseminated involvement of peripheral and central lymph node regions
M0	No evidence of the involvement of other extranodal sites
M1a	Noncontiguous involvement of tissue or organs external to the ocular adnexa (e.g., parotid glands, submandibular gland, lung, liver, spleen, kidney, breast)
M1b	Lymphomatous involvement of the bona marrow
M1c	Both M1a and M1b involvement

**Table 4 cancers-15-05782-t004:** Study with radiotherapy as first-line treatment.

Clinical Studies	Year	No. Patients	Median Age	Male/Female Ratio (M/F)	Histological	Initial Stage	Treatment	Rt Doses Gy (mean)	Median Follow Up (Years)	OS	LR	DR	CR
Stafford et al. [60]	2001	48	68 (35–94)	Unspecified	MALT (60%) CLL (23%) DLBCL 4% Other 13%	IEA (34), IIEA (6) IIIEA (2), IVEA (6)	EBRT	15 Gy–53.8 Gy (27.5 Gy)	5.4	100%	2%	12%	98%
Bhatia et al. [54]	2002	47	69 (32–89)	18/29	FL 25, MALT 8, DLBCL 12, Other 7	IAE	EBRT	low-grade tumors were 30 Gy (20–40.2); intermediate–high grade 40 Gy (30–51)	4.58	73.6%	2%	15%	100%
Uno et al. [61]	2003	50	61 (31–83)	33/17	MALT 48	I (100%)	EBRT	20–46 Gy Mean 36	3.8	91%	2%	6%	39%/ 76% (M/F)
Fung et al. [6]	2003	98	82 (11–95)	42/56	MALT (57%) Follicular (15%) DLBCL (9%) Other (12%)	I 65 III 5 IV 16	EBRT Photons e^−^	30.6 Gy	6.8	83%	2%	25%	96%
Cho et al. [31]	2003	68	46 (7–89)	31/37	MALT 61, DLBCL 2, MCL 2, other 3	I 51 II 8 III 4 IV 5		20–54 Gy	2.19	95%	12%	10%	68%
Goda et al. [43]	2011	89	56	37/52	MALT 100%	IE	Photons 93% e^−^ 3%	87 pts, 25 Gy 2pts, 36Gy	5.9	91% 7-y fu	9%	17%	99%
Fasola et al. [42]	2013	20	70 (38–88)	10/10	FL 11 MALT 8 Other 1	IE 7 IIE 3 IIIE 1 IV 9	EBRT	4 Gy (2 Gy × 2)	2.16	100%	4%	0%	85%
Ohga et al. [44]	2015	73	63 (22–90)	31/42	MALT 100%	IE	EBRT	30 Gy	3.8	100%	0%	18%	100%
Parikh et al. [64]	2015	79	59 (21–89)	28/51	MALT 75% Follicular (25%)	IE	EBRT	30.6 Gy	4.1	100%	0%	5.8%	100%
Woolf et al. [76]	2015	81	58.2 (22.6- 90.7)	37/44	MALT (88%) Follicular (6%) T-cell 1% B-precursor Lymphoblastic 1%	IE	EBRT	30–35 Gy	4.4	100%	0%	5%	100%
König et al. [28]	2016	CRT 45	64 years (range 24–84 years)	16/27	MALT 51.9% Follicular Lymphoma 13.5% immunocytomas (9.6%), OTHER B-cell lymphomas (25.0%)	I 43 II 4 III 2 IV 1 N/D 2	CRT	36 Gy (range 26–46 Gy)	11	100% at 2 years 85.6% at 5 years	CRT 11.6%5-yfu	CRT 10.1% 5-yfu	76.1%
		LDRT 7	Median age was 75 years (range 59–79 years)	3/4			LDRT	4 Gy (2 Gy × 2)	2	100% 2 y	2-year was 0%	31.40%- 2 year	
Pinnix et al. [65]	2017	22	64.5 (25–88)	12/10	MALT 64%; follicular 23%; MCL 9%; Other 4%	IE 15 IV 7	EBRT e^−^ 8 Photons 14 (1 pts systemic therapy with rituximab)	4 Gy (2 Gy × 2)	1.17	100%	4.5%	0%	86.4%
Kharod et al. [16]	2018	44	64 (10–88)	17/27	Unspecified	IAE 75% IIAE 25%	EBRT	25.5 Gy Mean (15–27.5)	4.9	89%	2%	20%	89%
Niwa et al. [77]	2020	81	66 (29–90)	42/39	MALT	IE	EBRT	30–36 Gy in 15–18 fractions	6.2	98.8%	0%	6.2%	69.1%
Xu et al. [57]	2021	32	56 (32–83)	21/11	MALT	IE	EBRT (IMRT) + electron beam with or without lens-sparing method	A: 22 pz 20 Gy/10 Fr EBRT+ 14 Gy/7 Fr Electron beam using lens shield B: 10 pz 32 Gy/16 Fr or 34 Gy/17 Fr without lens shield	7	100%	0%	9.4%	90.6%
Lee et al. [78]	2021	8	58 (35–65)	3/5	MALT	IE	EBRT	4 Gy (2 Gy × 2)	3.6	100%	0%	0%	75%
Leeson et al. [79]	2021	18	67 (44–87)	10/8	MALT	IE IIE	EBRT (3D-CRT or VMAT)	20–30 Gy with 1.5–2 Gy fraction sizes	2.8	100%	0%	16%	100%
Park et al. (2022) [21]	2022	14	60.5	7/7 1:1	MALT	IE	RT	4 Gy additional 24 Gy	2.35	90.6%	9.4%	0%	64.7%
Shelukar et al. [74]	2022	17	67 (24–80)	5/12	MALT 5 Other low-grade FL 4 Marginal zone lymphoma 3 MCL 1	I/II 16 III/IV 1	EBRT	4 Gy (2 Gy × 2)	3.25	100%	6%	18%	65%
de Castro et al. [75]	2022	7	75 (49–86)	unspecified	FL 3 MALT 2 Marginal zone 1 Low grade 1	I 4 II 2 IV 1	EBRT CHT (1 pts	4 Gy (2 Gy × 2)	1.8	57%	0%	14%	71%

Note: all the abbreviations in Table 4 are listed below. MALT—mucosa-associated lymphoid tissue; DLBCL—diffuse large B-cell lymphoma; MCL—mantle cell lymphoma; CRT—conventional radiotherapy, LDRT—low-dose radiotherapy; EBRT—external beam radiation therapy; VMAT—Volumetric modulated arc therapy; 3DCRT—three-dimensional conformal radiation therapy electrons; OS—overall survival; LR—local recurrence; DR—distal recurrence; CR—complete response; CHT—chemotherapy.

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
