# Peer review of "Radiotherapy of Orbital and Ocular Adnexa Lymphoma: Literature Review and University of Catania Experience"

_cancers, 2023, doi:10.3390/cancers15245782_

Round 1
Reviewer 1 Report
Comments and Suggestions for Authors
Excellent review. Line 143 needs reference.
Comments on the Quality of English LanguageMinor editing.
Author Response
|
Summary |
|
|
|
Thank you very much for taking the time to review our manuscript. Please find the detailed responses below and the corresponding corrections marked in red in the re-submitted files.
|
||
|
Point-by-point response to Comments and Suggestions for Authors |
|
|
|
Comments 1: Excellent review. Line 143 needs reference. |
||
|
Response 1: Thank you for your appreciation on our work. As pointed, we provide to add the missing citation.
|
||
|
Response to Comments on the Quality of English Language |
||
|
Comments 2: Minor editing. |
||
Response 2: We have, revised some of the highlighted parts. You will find them in red inside the revised submitted file.
Reviewer 2 Report
Comments and Suggestions for Authors
In this paper, the authors report a series of eleven patients affected by orbital and ocular adnexa lymphoma and propose a literature review of this topic.
The article is well written. Some reference links are missing.
In my opinion, the case series is small (11 patients) and heterogeneous. In fact, some of these patients were treated with proton beam, others by with external beam radiotherapy. Among these groups the treatment dose was different and some patients received systemic treatment and others didn’t.
Instead, the literature review is very interesting and readable, though it could be updated with most recent articles (i.e. Shelukar S, Fernandez C, Bas Z, Komarnicky L, Lally SE, Shields CL, Binder A, Porcu P, Alpdogan O, Martinez-Outschoorn U, Shi W. High local control and low ocular toxicity using ultra-low-dose "boom-boom" radiotherapy for indolent orbital lymphoma. Chin Clin Oncol. 2022 Dec;11(6):44. doi: 10.21037/cco-22-84. PMID: 36632978 or de Castro B, Peixeiro RP, Mariz JM, Oliveira Â. Ultra-low dose radiotherapy in the management of low-grade orbital lymphomas. Rep Pract Oncol Radiother. 2022 Jul 29;27(3):467-473. doi: 10.5603/RPOR.a2022.0044. PMID: 36186691; PMCID: PMC9518773..).
Author Response
|
1. Summary |
|
|
|
Thank you very much for taking the time to review this manuscript. Please find the detailed responses below and the corresponding revisions and corrections will be marked in red in the re-submitted file.
|
||
|
Point-by-point response to Comments and Suggestions for Authors |
|
|
|
Comments 1: In this paper, the authors report a series of eleven patients affected by orbital and ocular adnexa lymphoma and propose a literature review of this topic.
|
||
|
Response 1: Thank you for your appreciation on our work. As pointed, we provide to add the missing citation.
|
||
|
Comments 2: In my opinion, the case series is small (11 patients) and heterogeneous. In fact, some of these patients were treated with proton beam, others by with external beam radiotherapy. Among these groups the treatment dose was different and some patients received systemic treatment and others didn’t. |
||
|
Response 2: We are aware of the small sample size, but since it is a fairly rare pathology, we performed a retrospective research from the clinical data in our possession over a large period of time (2010-2021). Patients who received systemic therapy in addition to radiotherapy came to our center for consultation only following the start of the aforementioned therapy. As regards the heterogeneity of the dose and the methods, we wanted to report them to emphasize the non-univocal consensus that the literature has in common. In our center we have the possibility to treat not only with external beams, but also with protons, not commonly used elsewhere.
Comments 3 Instead, the literature review is very interesting and readable, though it could be updated with most recent articles (i.e. Shelukar S, Fernandez C, Bas Z, Komarnicky L, Lally SE, Shields CL, Binder A, Porcu P, Alpdogan O, Martinez-Outschoorn U, Shi W. High local control and low ocular toxicity using ultra-low-dose "boom-boom" radiotherapy for indolent orbital lymphoma. Chin Clin Oncol. 2022 Dec;11(6):44. doi: 10.21037/cco-22-84. PMID: 36632978 or de Castro B, Peixeiro RP, Mariz JM, Oliveira Â. Ultra-low dose radiotherapy in the management of low-grade orbital lymphomas. Rep Pract Oncol Radiother. 2022 Jul 29;27(3):467-473. doi: 10.5603/RPOR.a2022.0044. PMID: 36186691; PMCID: PMC 9518773..).
Response 3: We thank you for the precious advice you gave us, to enrich the bibliography with the studies you kindly indicated. You will find them on Table 4 and at the end of Discussion chapter. |
||
Reviewer 3 Report
Comments and Suggestions for Authors
This article reports their clinical experience of radiotherapy on 11 patients with orbital and ocular adnexa lymphomas (5 eyelid, 3 conjunctiva, and 3 lacrimal glands) and review literature.
The major concern is
1. Limited case number in their study group with 3 different location
2. No pathological diagnosis of lymphoma type on their study patients or literature review (B cells or T-cell lymphoma, Hodgkin or non-Hodgkin tumors, REAL classification? : Mucosa-associated lymphoid tissue (MALT) lymphomas, or Chronic lymphocytic lymphoma (CLL) or Follicular center lymphoma, or High-grade lymphomas such as large cell lymphoma, lymphoblastic lymphoma, and Burkitt lymphoma.)
All these factors may influence the result and outcome of orbital radiotherapy which may this review article less significant.
Author Response
|
1. Summary |
|
|
|
Thank you very much for taking the time to review this manuscript. Please find the detailed responses below. |
||
|
3. Point-by-point response to Comments and Suggestions for Authors
|
|
|
|
Comments 1: This article reports their clinical experience of radiotherapy on 11 patients with orbital and ocular adnexa lymphomas (5 eyelid, 3 conjunctiva, and 3 lacrimal glands) and review literature. The major concern is 1. Limited case number in their study group with 3 different location. |
||
|
Response 1: Thank you for pointing this out. Since it is a rather infrequent neoplasm, we have performed a retrospective search of the clinical information in our archives. We are aware that this is not a large sample, but we believe it is a sample consistent with other articles cited in our manuscript (e.g. Lee et al or Park et al). However, we wanted to report our experience since we can treat patients with both external beams and protons, which are not very widespread diffused.
|
||
|
Comments 2: 2. No pathological diagnosis of lymphoma type on their study patients or literature review (B cells or T-cell lymphoma, Hodgkin or non-Hodgkin tumors, REAL classification?: Mucosa-associated lymphoid tissue (MALT) lymphomas, or Chronic lymphocytic lymphoma (CLL) or Follicular center lymphoma, or High-grade lymphomas such as large cell lymphoma, lymphoblastic lymphoma, and Burkitt lymphoma.) |
||
|
Response 2: We Agree with you about this missing information, and we thank you for this suggestion. We have, revised our manuscript and added the missing data. You can find it in Table 1 and also in the Capter “Materials and Methods”. |
||
Reviewer 4 Report
Comments and Suggestions for Authors
In the current manuscript, the authors summarized the utility of radiation therapies on cases with orbital and ocular adnexa lymphoma (OOAL). Although the manuscript contains several information that will be informative for the readers, the manuscript needs to be more structured so that the readers can easily understand the contents.
1. In the Introduction, it is better to summarize what the authors want to do, and why the authors write this review article. The current section of Introduction seems just show the focus on OOAL, and this is not directly related to the contents of this manuscript.
2. The authors summarized their own cases in Table 1, but the summary seems not informative for the readers. This is partially because the authors do not summarize well in the manuscript and the discussion on the cases from the literature review will be need.
3. As the title of manuscript suggests the utility of radiation therapy for OOAL cases, the authors do not discuss the utility enough in the manuscript. It would be better to summarize more focusing on who needs the radiation therapy or how the clinicians can manage the cases.
4. It is hard to understand the Table 4. For example, it is necessary to summarize information in an easy-to-understand format, such as which groups are likely to benefit from radiation therapy.
5. Some citations are missing as errors in the manuscript.
Author Response
|
1. Summary |
|
|
|
Thank you very much for taking the time to review this manuscript. Please find the detailed responses below and the corresponding revisions and changes in the re-submitted files. |
||
|
3. Point-by-point response to Comments and Suggestions for Authors |
|
|
|
Comments 1: In the Introduction, it is better to summarize what the authors want to do, and why the authors write this review article. The current section of Introduction seems just show the focus on OOAL, and this is not directly related to the contents of this manuscript.
|
||
|
Response 1: Thanks for pointing this out. We have taken steps to specify the intent of our work. Our intent was to clarify the central role of radiotherapy and to enrich it with our clinical experience. You will find the changes within the introductory chapter. |
||
|
Comments 2: The authors summarized their own cases in Table 1, but the summary seems not informative for the readers. This is partially because the authors do not summarize well in the manuscript and the discussion on the cases from the literature review will be need. |
||
|
Response 2: We Agree with your comment, and made some changes to Table 1 which summarizes our sample. We have also added a more detailed description of the sample under examination in Materials and methods. |
||
Comments 3: As the title of manuscript suggests the utility of radiation therapy for OOAL cases, the authors do not discuss the utility enough in the manuscript. It would be better to summarize more focusing on who needs the radiation therapy or how the clinicians can manage the cases.
Response 3: Thank you for this important comment. We have further emphasized the role of radiotherapy and how the clinician can best use this therapeutic option in daily clinical practice.
We added this focus inside “Discussion” Chapter.
Comments 4: It is hard to understand the Table 4. For example, it is necessary to summarize information in an easy-to-understand format, such as which groups are likely to benefit from radiation therapy.
Response 4: Given the amount of information included within the table, we made a change in layout and preferred to better clarify the intent of the aforementioned table within the manuscript.
Comments 5: Some citations are missing as errors in the manuscript.
Response 5: We have taken care of fixing the missing references. you will find them in red in the text.
Round 2
Reviewer 3 Report
Comments and Suggestions for Authors
Lymphomas are not infrequent Orbital and Ocular adnexa neoplasms as compred to other orbital and ocular tumors. Limited cases with different type lymoma among different location make this study add no new information.
Author Response
Thank you very much for taking the time to review our manuscript
Comment 1: Lymphomas are not infrequent Orbital and Ocular adnexa neoplasms as compred to other orbital and ocular tumors. Limited cases with different type lymoma among different location make this study add no new information.
Response 1: We appreciate your comments and suggestions. However, from the literature data collected during the formulation of our manuscript, many other authors among those cited report the localization of lymphoma in the periorbital region as less frequent than in other body areas, even if it is an extranodal localization affected by this pathology. Nonetheless, it presents clinical difficulties in the differential diagnosis with other pathologies of the same body area. We are aware of the limited number of patients (like other works in the literature), but we have reported our experience by referring to various therapeutic strategies (EBRT, PBT, combined therapy) and dose fractionations.
Reviewer 4 Report
Comments and Suggestions for Authors
The authors have responded to the reviewers comments and have modified the manuscript. The changes have made the manuscript clearer.
Author Response
Thank you very much for taking the time to review this manuscript.
Comment 1: The authors have responded to the reviewers comments and have modified the manuscript. The changes have made the manuscript clearer.
Response 1: We thank you for your valuable advice and are pleased that this has improved the usability of the manuscript.